# Middle east warming in spring enhances summer rainfall over Pakistan

**Baosheng Li** [1,2], **Lei Zhou** [2,3] ✉, **Jianhuang Qin**[2,4], **Tianjun Zhou** [5], **Dake Chen**[1,2,3], **Shugui Hou**[3] & **Raghu Murtugudde** [6,7] ✉

The edge of a monsoon region is usually highly sensitive to climate change. Pakistan, which is located on the northern edge of the Indian monsoon, is highly vulnerable to heavy rainfall and has witnessed several debilitating floods exacerbated by global warming in recent years. However, the mechanisms for the frequent Pakistan floods are yet not fully understood. Here, we show that the Middle East is undergoing an increase in land heating during spring, which is responsible for 46% of the intensified rainfall over Pakistan and northwestern India during 1979–2022. This springtime land warming causes a decline in sea level pressure (SLP), which strengthens the meridional SLP gradient between the Middle East and the southern Arabian Sea and drives the changes of low-level jet (LLJ) subsequently. The impact persists into summer and results in a northward shift of the monsoonal LLJ, accompanied by strong positive vorticity in the atmosphere and enhanced moisture supply to Pakistan. Consequently, the transition region between the summer monsoon in South Asia and the desert climate in West Asia is shifted northwestward, posing significantly enhanced risk of floods over Pakistan and northwestern India.

In the past four decades, record-breaking floods have been reported over Pakistan and northwestern India frequently[1,2]. Climatologically, Pakistan is an arid to semi-arid region (Supplementary Fig. 1). Nevertheless, it has experienced flooding every year since 2010 as reported by the Ministry of Water Resources of Pakistan. In 2010, Pakistan suffered an unprecedented flooding due to heavy monsoon rainfall. Twenty million people were affected, and the death toll was nearly 2000[3]. In 2022, the devastating flood in Pakistan caused by abnormal summer rainfall became the worst natural disaster in the country's history[4]. According to the Pakistan National Disaster Management Authority, the flood left one-third of the country submerged, at least 1739 dead, and 8 million displaced[5]. The damages and economic losses were over 30 billion U.S. Dollars (Pakistan's GDP was 310 billion U.S. Dollars in 2022) and the reconstruction required more than 16 billion

U.S. Dollars. This event is a poster-child for climate extremes, climate change impacts, and consequences of climate vulnerability. The satellite imagery reveals a 20–50% increase in the proportion of the population exposed to floods from 2000 to 2015 in Pakistan, and the proportion is expected to continue to increase out to 2030[6]. Since 1979, the summer monsoon rainfall has increased by 46% over Pakistan and northwestern India (black box in Fig. 1a; see *Methods*), with a significantly increasing trend of 0.4–0.6 mm day⁻¹decade⁻¹. The rainfall increases over Pakistan are pronounced and robust among different satellite products (Supplementary Fig. 2). Thus, there is an urgent need to understand the physical mechanism that is responsible for the debilitating heavy rainfall events over Pakistan.

Moisture supply for the summer rainfall along the west coast of the Indian subcontinent is mainly transported from the Arabian Sea by

[1]State Key Laboratory of Satellite Ocean Environment Dynamics, Second Institute of Oceanography, Ministry of Natural Resources, Hangzhou, China. [2]Southern Marine Science and Engineering Guangdong Laboratory (Zhuhai), Zhuhai, China. [3]School of Oceanography, Shanghai Jiao Tong University, Shanghai, China. [4]College of Oceanography, Hohai University, Nanjing, China. [5]State Key Laboratory of Numerical Modeling for Atmospheric Sciences and Geophysical Fluid Dynamics, Institute of Atmospheric Physics, Chinese Academy of Sciences, Beijing, China. [6]Indian Institute of Technology Bombay, Mumbai, India. [7]University of Maryland, College Park, MD, USA. ✉e-mail: zhoulei1588@sjtu.edu.cn; mahatma@umd.edu

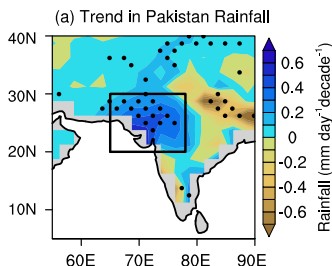
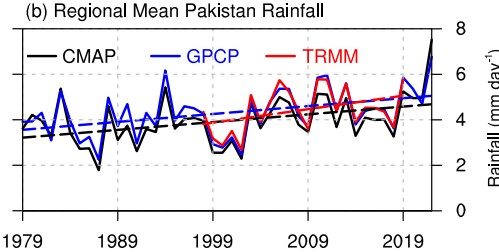

**Fig. 1 | Summer rainfall increases in Pakistan. a** The trend in summer rainfall (shading; unit: mm day$^{-1}$ decade$^{-1}$) for 1979–2022 in GPCP dataset. The dotted areas are significant at the 95% confidence level using the Mann–Kendall test. **b** Index of the averaged summer rainfall over land within the region of 20°–30°N and

65°–78°E, which is marked with the black rectangle in **a**. Data are from CMAP (black line), GPCP (blue line), and TRMM (red line). The colored dashed lines denote the corresponding linear trends in the three rainfall products, and they are obtained by the least squares regression.

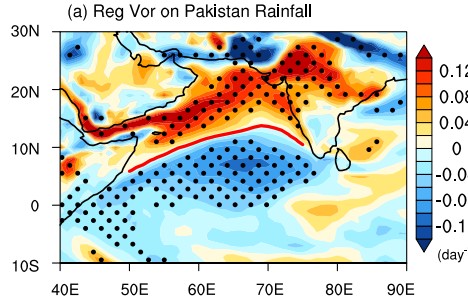
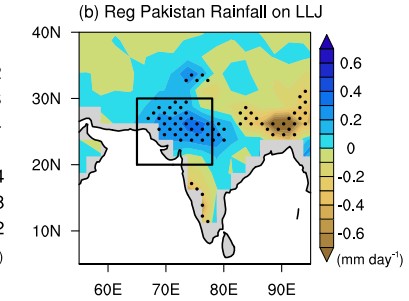

**Fig. 2 | Low-Level Jet leads to summer rainfall over Pakistan. a** Regression coefficients of the absolute vorticity (shading; unit: day$^{-1}$) onto the normalized Pakistan's summer rainfall index for all summers from 1979 to 2022. The locations where the regressed vorticity is zero are marked with a red line. The red line represents the position of Low-level jet (LLJ) maxima in summer. **b** The regression

of rainfall amount derived from GPCP dataset (shading; unit: mm day$^{-1}$) onto the normalized LLJ location index in summer during the period of 1979–2022. The dotted areas in both **a** and **b** are statistically significant at the 95% confidence level using the Students' $t$ test.

the low-level jet (LLJ)[7,8]. The LLJ is sustained by the land-sea thermal contrast around the Indian Ocean, with the orography of the East African highlands helping to channel the cross-equatorial flow into a jet[9,10]. That results in the strong southwesterly winds from the ocean to the land. The southwesterly winds associated with LLJ that sweep across the Arabian Sea onto the Indian subcontinent are a quintessential signature of the Indian summer monsoon (ISM)[11]. The LLJ also feeds on the cross-equatorial southerly winds, which transport moisture and support the convection over the entire Indian monsoon region. It accounts for nearly half of the global interhemispheric transport of air in the lower troposphere during ISM[12,13].

In this study, we show that the enhanced Middle East warming under global warming leads to a persistent poleward shift of LLJ and thrusts the moisture supplies northward, bringing unprecedented rainfall to Pakistan. The Middle East warming is almost two times faster than other inhabited parts of the world[14,15]. The consequent poleward shift of the LLJ over the Arabian Sea causes a dramatic increase in atmospheric instability and rainfall over Pakistan and northwestern India. Unlike the wet-get-wetter paradigm[16,17], the mechanism proposed here tends to turn the arid region into a rainy one. However, the observed increase in the devastating rainfall over Pakistan is largely missed by the historical climate simulations of most CMIP6 models. Such inadequate warning will leave millions of people unprepared in terms of adaptation to global warming risks.

## Results

### Northward shift of low-level jet induces Pakistan's rainfall under global warming

Over Pakistan and nearby northwestern India (black box in Fig. 1; 20°N-30°N; 65°E-78°E), the regional rainfall has increased by 46% or equivalent to 0.37 mm day$^{-1}$ decade$^{-1}$ (Lines in Fig. 1b) from 1979 to 2022. The regressed vorticity anomaly onto the Pakistan's summer

rainfall index displays the typical characteristics of LLJ[18], since LLJ is defined as the line with zero absolute vorticity at 850 hPa (see *Methods*). Positive (negative) vorticity lies over northern (southern) Arabian Sea (Fig. 2a). The climatological LLJ location (see *Methods*) is around 10°N over the Arabian Sea with positive vorticity just to the north over the ocean and western-southwestern India (Supplementary Fig. 3a). The regressed positive vorticity anomaly extends into northwestern India and Pakistan, which shares a similar spatial pattern with the trends in LLJ (Supplementary Fig. 3b). Since 1979, the LLJ has shown a northward shift over the Arabian Sea as previously reported[19,20]. The northernmost latitudinal location is shifted to approximately 18°N off western India. In addition, the trend in the LLJ location index (see *Methods*) is approximately 0.27° latitudes per decade from 1979 to 2022 (red bars and the red line in Fig. 3c). Correspondingly, the regressed summer rainfall anomaly onto the LLJ location index displays a pronounced increase over Pakistan and northwestern India (Fig. 2b). This suggests that the LLJ shift is vital to rainfall enhancement over Pakistan.

### Middle East heating in spring leads to northward shift of Low-Level Jet

To understand the mechanism of the northward shift of LLJ, we regress the spring land skin temperature onto the summer LLJ location index and find significant positive changes over the Middle East (30°N-50°N; 45°E-70°E; Fig. 3a). From 1979 to 2022, the spring heating over the Middle East has a significant increasing trend (Fig. 3b). It implies that the intensified surface heating over the Middle East has pulled the LLJ northward. We employ the mean skin temperature in spring within the Middle East as the land heating index, which displays a notably increasing trend of 0.5 °C decade$^{-1}$ (blue bars and the blue line in Fig. 3c). We examine the scatter plot between the land heating over the Middle East in spring and the LLJ location in summer in past 44 years

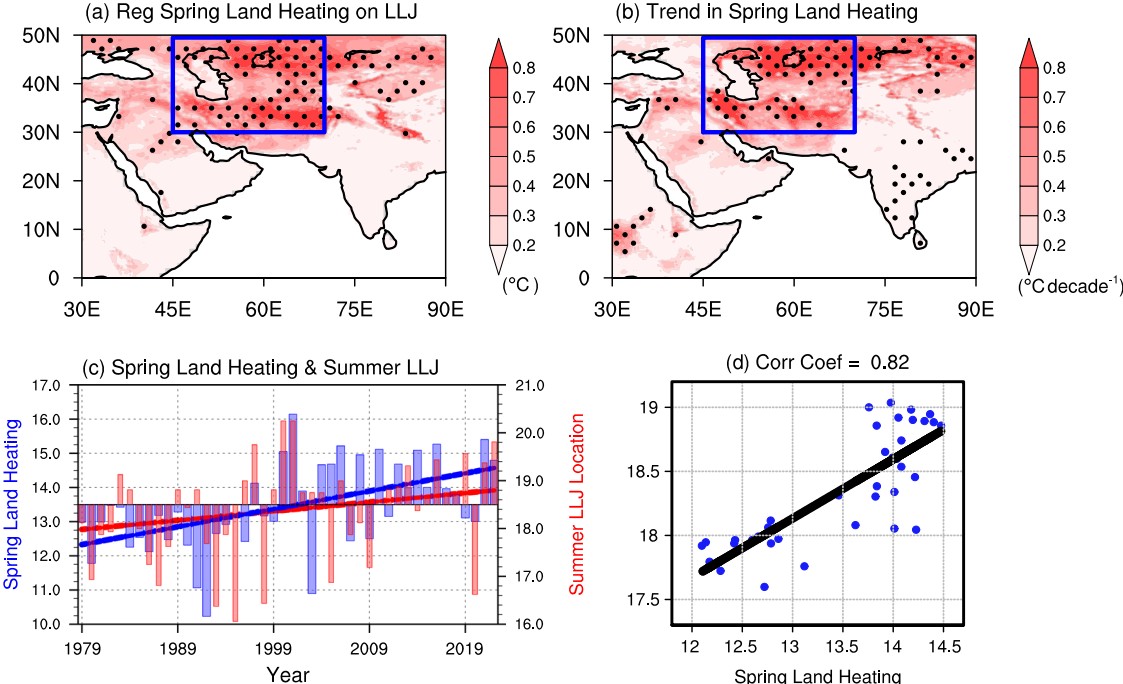

**Fig. 3 | Warming trend over the Middle East leads to northward shift of LLJ.**
**a** Regression coefficients of the spring skin temperature (unit: °C) onto the normalized summer LLJ location index during the period of 1979–2022. The dotted areas are significant at the 95% confidence level using the Student's *t* test. **b** Trends in spring skin temperatures (unit: °C decade⁻¹) from 1979 to 2022. The dotted areas are significant at the 95% confidence level using the Mann–Kendall test. **c** Indices of regional mean spring land heating over the Middle East (blue bars; unit: °C) and summer LLJ location over the Arabian Sea (red bars; unit: degree). Their linear trends (solid lines) are obtained by the least squares regression. **d** The scatter plot between spring land heating index (x-axis; unit: °C) and summer LLJ location (y-axis; unit: degree) in each year from 1979 to 2022 and the time series are run through a 7-year moving average. The black line shows the least squares regression of the scattered dots.

from 1979 to 2022 (Fig. 3d). The correlation coefficient between these two components is 0.82, which is statistically significant at the 99% confidence level. The significant relationship indicates that a warmer Middle East during spring is favorable for the northward shift of summer LLJ. In summary, the northward shift of LLJ due to the continuous Middle East heating in spring causes heavy summer rainfall over Pakistan and northwestern India.

## Dynamical bounds between Middle East warming and increasing rainfall in Pakistan

Under global warming, the Middle East heating pulls the entire dynamical system responsible for the excess Pakistan's rainfall via LLJ northward. During boreal spring, a large-scale meridional surface temperature gradient develops due to the different heat capacities of the land and the adjoining ocean through the seasonal cycle of solar radiation[21]. Due to the net effect of anthropogenic forcing[22,23], the Middle East is warming faster during spring starting around 1979, as shown by the land heating differences between the earlier period (1979–2000) and the recent period (2000–2022) (Fig. 4a). The maximum temperature difference is 2 °C (shading over land in Fig. 4a). As a result of the land warming in spring, the sea level pressure (SLP) over the nearby landmasses to the north of the Arabian Sea reduces by -120–200 Pa by increasing the sensible heat fluxes (contours in Fig. 4a; Supplementary Fig. 4a), which enhances the northward pressure gradient force from the southern to the northern Arabian Sea. This meridional pressure gradient drives geostrophic winds in the lower troposphere along isobars (vectors in Fig. 4a) resulting in a weak cyclonic gyre at 10°N over the Arabian Sea (shading over sea in Fig. 4a).

The land heating persists from spring to early summer. Its impacts on SLP, horizontal winds, and low-level vorticities also persist coherently until the southwesterly wind anomalies penetrate far to the north of 20°N (vectors in Fig. 4b). Consequently, the LLJ lies over the Arabian

Sea (red contour in Fig. 4b) as a cyclonic (anticyclonic) gyre is reinforced over northern (southern) Arabian Sea (shading over sea in Fig. 4b; Supplementary Fig. 4b). Consistently, we see pronounced kinetic energy (Supplementary Fig. 4c) in the western Indian Ocean and prevailing southwesterly winds over the Arabian Sea. As a result, the cyclonic gyre associated with LLJ in summer is significantly strengthened to the north of the Arabian Sea, in particular over Pakistan (Fig. 4c). The moisture instability and atmospheric barotropic instability are both increased over Pakistan due to the northward shifted LLJ (Supplementary Fig. 5). Consequently, the regions become unstable and favorable for deep convection[24,25]. In addition, the LLJ is accompanied by strong westerly winds and a pronounced positive vorticity, which provide a track for the invasion of the low-pressure systems into Pakistan, and transport moisture from the BoB to further nourish the floods[26,27]. Meanwhile, vertically integrated moisture transport (VIMT; shading over sea in Fig. 4d; see *Methods*) yields distinct positive moisture anomalies over Pakistan. By decomposing the VIMT, the southwest direction of VIMT is evidenced (vectors in Fig. 4d) and mainly corresponds to the LLJ. It indicates that the intensified moisture supply is largely transported from the Arabian Sea by the LLJ to Pakistan and northwestern India in the recent period of 2000–2022. As a consequence, the summer rainfall (shading over land in Figs. 2b and 4d) is intensified over Pakistan and northwestern India, which is roughly confined to 20°N–30°N and 65°E–78°E. In summary, the sustained effects of spring land heating push the LLJ to the north, leading to an increase in vorticity and moisture supply to northwestern India and Pakistan, which eventually leads to enhanced rainfall over these regions during summer.

## Discussion

Pakistan is located on the northern edge of the Indian monsoon region. It is particularly vulnerable to the shift of monsoon climate, compared

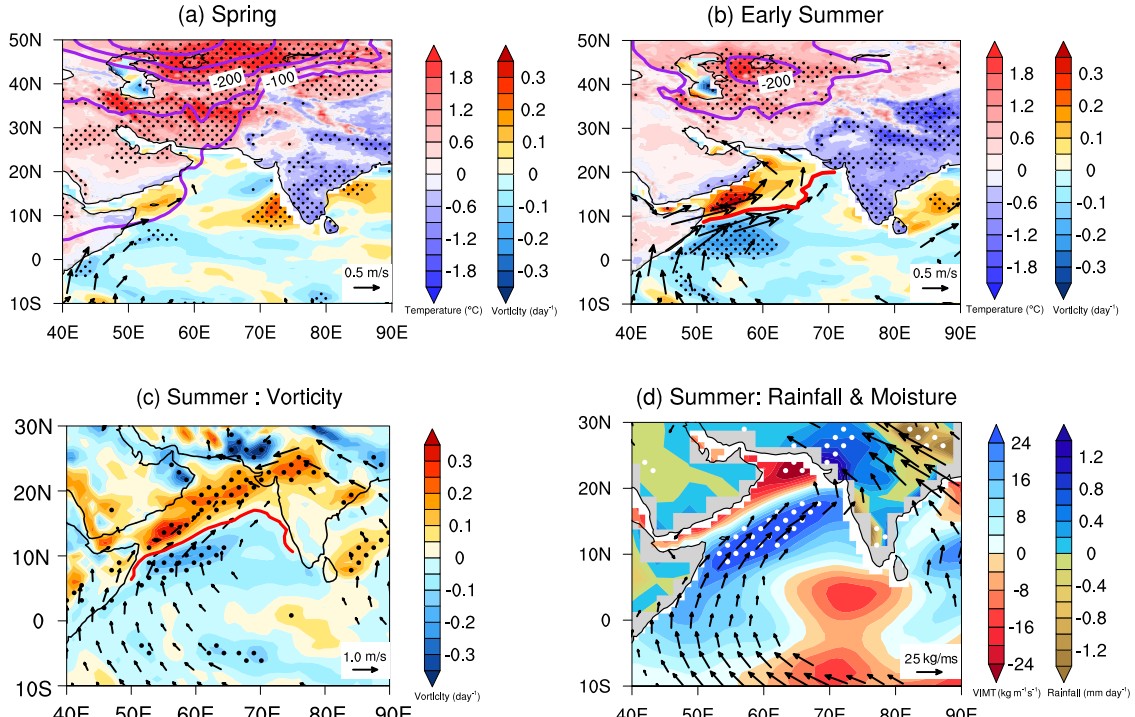

**Fig. 4 | Dynamical processes from Middle East heating in spring to summer rainfall in Pakistan.** Differences in the land heating (shading over land; unit: °C), the sea level pressure (contours; unit: Pa), the absolute vorticity (shading over sea; unit: day$^{-1}$) and horizontal winds (vectors; unit: m s$^{-1}$) at 850 hPa in (**a**) spring (March–May) and (**b**) early summer (May–June) between the recent period (2001-2022) and the earlier period (1979-2000). **c** is the same as **a** but for the absolute vorticity (shading; unit: day$^{-1}$) and horizontal winds (vectors; unit: m s$^{-1}$) at 850 hPa in summer (June–September). **d** same as **a** but for the vertically integrated moisture transports (VIMT; shading over sea; unit: kg m$^{-1}$ s$^{-1}$) and their zonal and meridional components (vectors; unit: kg m$^{-1}$ s$^{-1}$), and summer rainfall (shading over land; unit: mm day$^{-1}$). Dotted areas are significant at the 95% confidence level using the Student's *t* test.

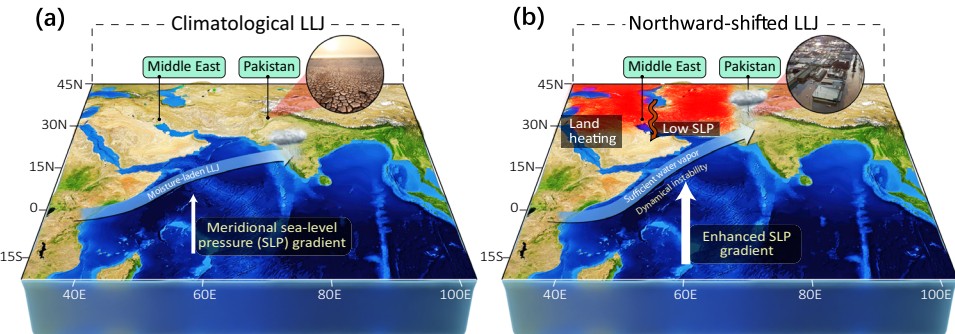

**Fig. 5 | Sketch for the impact of the spring Middle East warming on summer rainfall over Pakistan. a** Stands for the climatological situation, and **b** shows the processes associated with the northward-shifted LLJ under global warming. In Fig. 5a, Pakistan and northwestern India are arid to semi-arid regions. The thin white arrow denotes the weak meridional sea level pressure (SLP) gradient. In contrast, in Fig. 5b, the red color over the Middle East represents the warming in spring. The thick white arrow denotes the enhanced meridional SLP gradient. The LLJ shifts northward compared to the climatology and brings more rainfall to Pakistan and northwestern India.

to the core monsoon region[28]. This study advances our understanding of the observed summer rainfall changes in Pakistan from the perspective of spring land heating and LLJ. The entire process originating from the Middle East warming is illustrated in Fig. 5. The Indian summer monsoon region expands and pushes the transition region northward under global warming. Dynamically, the remote land heating, when interacting with the atmospheric LLJ, leads to a wetter climate over the arid region by enhancing moisture transport. Such a land-air interaction mechanism is different from the well-established concept of rainfall increasing in the presently wet regions; the so-called wet gets wetter mechanism[16,17] states that the abundant moist supply strengthens local atmospheric moisture convergence. The close

relationship between the precursors in terms of the spring land warming and LLJ points to the potential predictability of the changes in summer rainfall in a warming scenario. This should also provide a pathway to improve simulations for summer rainfall from the relationship between spring warming and LLJ. This demands special attention to the future of the Middle East warming and its implications of heavy rainfall over the highly vulnerable regions in the Indian subcontinent.

Future changes in Pakistan's summer rainfall are projected to increase under three Shared Socioeconomic Pathway 2-4.5 (SSP2-4.5), SSP3-7.0 and SSP5-8.5 scenarios (Supplementary Table 1). The LLJ also displays a robust northward shift and it plays an important role in

Pakistan's rainfall under all future scenarios. Thereby, in light of the crippling flooding in Pakistan, the trend in summer rainfall and its relationship with remote spring land heating over the Middle East assume significance in the projected future warming scenarios.

Both anthropogenic external forcings and internal variability can affect monsoon changes[29,30]. The multi-model ensemble mean (MMEM) of 192 members from the CMIP6 estimates that the external forcing and internal variability contribute to 81% and 19% of the warming trend over the Middle East, respectively (Supplementary Fig. 6). The rainfall trend in the MMEM denotes that the external forcing has a 36% contribution to the observed rainfall trend, and the internal variability is responsible for the rest. Nonetheless, we acknowledge that the separation of the contributions of external forcing and internal variability could be affected by model performances. We find that CMIP6 models show weaknesses in the simulation of both spring warming tendency and the summer rainfall changes. For example, the ERA5 features a spring land warming of 0.54 °C decade$^{-1}$ over the time period of 1979–2014 (Supplementary Fig. 7). The magnitude of the warming reproduced by CMIP6 models is only about 0.2 °C decade$^{-1}$. The observed rainfall trend over Pakistan is about 0.28 mm day$^{-1}$ decade$^{-1}$ in GPCP data. The corresponding changes in CMIP6 models are less than 0.2 mm day$^{-1}$ decade$^{-1}$. Hence, we highlight that there is an urgent need to improve model simulations of rainfall over Pakistan. It would be unfortunate if the inconsistency in simulating the observed trends of summer rainfall results in the neglect of the potential risk of flooding in the vulnerable region of Pakistan in the coming years and decades which may lead to further widening gap in adaptation.

## Methods

### Datasets
Monthly atmospheric horizontal and vertical winds, specific humidity, sea level pressure, and skin temperature are obtained from ERA5 reanalysis provided by the European Center for Medium-Range Weather Forecasts[31]. The period is from 1979 to 2022 and the spatial resolution is 0.25° latitude × 0.25° longitude. The reanalysis products from NCEP-NCAR[32] are also employed to ensure the robustness of the results, which are not shown in the main text because the results are qualitatively the same as ERA5. Monthly precipitation data are obtained from the Global Precipitation Climatology Project (GPCP)[33] and the CPC Merged Analysis of Precipitation (CMAP)[34], with a horizontal resolution of 2.5°. Monthly precipitation data with a horizontal resolution of 0.25° from the Tropical Rainfall Measuring Mission (TRMM)[35] are also employed in the analyses. The skin temperatures are used to represent land heating. The spring season is defined as March to May (MAM), and the summer season corresponds to the monsoon, i.e., June to September (JJAS). The early summer season corresponds to May–June (MJ). All analyses using observations are conducted for the period from 1979 to 2022.

Twenty coupled atmosphere-ocean general circulation models (CGCMs) of the sixth phase of Coupled Model Inter-comparison Project (CMIP6) are used (Supplementary Table 2). Monthly outputs of surface temperature, rainfall and horizontal winds are obtained from the selected models, including the historical simulations and the projections under three scenarios, namely the Shared Socio-economic Pathway 2-4.5 (SSP2-4.5), SSP3-7.0, and SSP5-8.5. To compare with observations over the same period, the outputs for the period of 1979–2014 are selected. The resulting fields from each model are interpolated to a grid of 2.5° latitude ×2.5° longitude to calculate the multi-model ensemble mean (MMEM). To clarify the relative impacts of external forcing and internal variability on the rainfall trend, the MMEM of 192 model members from the CMIP6 is used to estimate the contribution of external forcing, whereas the contribution of internal variability is estimated by the difference between the observed trend and the MMEM trend. The uncertainty is

evaluated with one standard deviation of the trend in each member. For more details about the CMIP6 models, see https://esgf-node.llnl.gov/projects/cmip6/.

### LLJ definition
The LLJ maxima are considered as the zero absolute wind vorticity anomalies at 850 hPa[18]. Accordingly, positive and negative absolute vorticities are observed on the northern and the southern sides of LLJ, respectively. Therefore, the location of the LLJ maximum, that is zero absolute vorticity, is taken as the proxy for the LLJ location. The LLJ shift is determined by the change in the zonally averaged LLJ location over the Arabian Sea using the location of absolute vorticity being zero. As the pronounced LLJ shift occurs over the eastern Arabian Sea, the LLJ location index is determined by the zonal mean between 60°E and 75°E within 5°N and 25°N. It is consistent with the distinct changes in positive vorticity anomalies over Pakistan and nearby regions.

### Trend and its statistical significance
The Sen's slope method is applied to obtain linear trends, which is the most popular non-parametric technique for determining linear trends[36]. Accordingly, the trends in summer rainfall (Fig. 1), spring land heating (Fig. 3b), and summer LLJ (Supplementary Fig. 3b) are determined from 1979 to 2022 by the Sen's slope method. Similarly, the simulated trends in summer rainfall over Pakistan and spring land heating over the Middle East are also obtained by the Sen's slope method during 1979–2014. The Mann–Kendall test is applied to evaluate the 95% significance level of the linear trends. In addition, changes in variables are also explored by dividing the study period of 1979–2022 into two subperiods: the earlier period of 1979–2000 and the recent period of 2001–2022. The composite differences are used to explore the changes in each variable (Fig. 4), and the significance is determined using a standard two-tailed Student's $t$-test. Moreover, the Student's $t$ test is also applied to assess the significance of regression analyses.

### Moisture transport
The vertically integrated moisture transport (VIMT) is used to measure the moisture supply. Following Chen[37], Godfred-Spenning and Reason[38], the VIMT is calculated by

$$VIMT = \frac{1}{g} \int_{ps}^{ptop} q\mathbf{V}dp, \tag{1}$$

where $\mathbf{V} = (u, v)$ and $u$ and $v$ are the zonal and meridional winds; $q$ is the specific humidity; $g$ is the acceleration due to gravity; $ps$ and $ptop$ are 1000 hPa and 300 hPa, respectively. In the tropics, the specific humidity above 300 hPa is at least two orders of magnitude smaller than near the surface, and the moisture transports therefore contribute little to the total VIMT[39]. In this way, the zonal and meridional components of VIMT are calculated by $\frac{1}{g}\int_{ps}^{ptop} qu\,dp$ and $\frac{1}{g}\int_{ps}^{ptop} qv\,dp$, respectively.

## Data availability
The data that support the findings of this study are freely available. The ERA5 reanalysis products are provided by the European Centre for Medium-range Weather Forecast (ECMWF), which can be obtained from their website at https://cds.climate.copernicus.eu/. The NCEP/NCAR reanalysis product, the GPCP and the CMAP datasets are all publicly available from the National Oceanic and Atmospheric Administration (NOAA; https://psl.noaa.gov/data/gridded/). The TRMM dataset is available at https://disc.gsfc.nasa.gov/datasets/TRMM_3B42_7/. The CGCM outputs from CMIP6 are produced by the World Climate Research Programme's Working Group at https://esgf-node.llnl.gov/projects/cmip6/.

## Code availability

The data in this study were analyzed with NCAR Command Language (NCL; http://www.ncl.ucar.edu/). All relevant codes used in this study are available, upon request, from the corresponding author.

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

## Acknowledgements

B.S.L. thank the grant (42206023) from the National Natural Science Foundation. L.Z. acknowledges support from the National Natural Science Foundation of China (42125601, 42076001). R.M. gratefully

acknowledges the Visiting Faculty position at the Indian Institute of Technology, Mumbai.

## Author contributions
B.S.L., L.Z., and R.M. designed the study. B.S.L. performed the data analysis and prepared all figures. B.S.L. and L.Z. wrote the paper. R.M., J.H.Q., T.J.Z., S.G.H. and D.K.C. contributed to the interpretation of the results and the improvement of the manuscript.

## Competing interests
The authors declare no competing interests.
