## [Peer Review File · Nature Communications]

Middle East Warming in Spring Enhances Summer Rainfall over PakistanReviewers' comments:

Reviewer #1 (Remarks to the Author):

Dear Authors,

I appreciate the efforts taken by the authors to find out the reason for the frequent Pakistan flood. However, the present study failed in citing some relevant works and lacking in-depth analysis.

Please find my comments and suggestions below to improve the article

1. The present study links the middle east land warming to the increasing rainfall over Pakistan and north-west India, which can intensify the floods over Pakistan. The proposed mechanism is the land warming causes a reduction in the surface pressure, which enhances the meridional pressure gradient between the Middle East and the Southern Arabian Sea, causing a poleward shift in the location of the LLJ. The northward shifted LLJ enhances the atmospheric vorticity and moisture supply to north-west India as well as Pakistan and thereby enhances the monsoon rainfall. However, a similar mechanism is already explained by Sandeep and Ajayamohan, (2015). How the mechanism explained in the present study differ from Sandeep and Ajayamohan's, (2015) mechanism?
2. The location and intensity of LLJ are influenced by the deep convection over India and nearby oceans (Joseph and Sijikumar, 2004; Sijikumar and Aneesh, 2022). What is the role played by the recent changes in the convection pattern over the Indian land region and the nearby oceanic parts in the observed northward propagation of LLJ?
3. According to Otto et al., 2023, the flood during the monsoon season in Pakistan is mainly caused by the advection of moisture from the Bay of Bengal and the Arabian Sea, associated with the dynamic features in the atmosphere, such as the mid-latitude jet stream (E.g.- 2010 flood). The floods are also caused by the movement of monsoonal depressions originating in the Bay of Bengal and travelling over the Indo-Gangetic plains or central India, thereby causing high rainfall over Pakistan (E.g.- 2022 flood). Thus, the root cause of the floods in Pakistan is the episodes of heavy rainfall events caused by the above-mentioned mechanisms. How the formation of depressions and the mid-latitude jet streams have changed due to the middle east land warming and how has it influenced the flood over Pakistan?
4. Is the regression coefficient calculated after removing the linear trend from the variable?
If regression coefficients are not calculated after removing the linear trend, then replot all the figures showing the regression coefficients after detrending the variables.
5. Line 129: How surface warming can cause ~120-200 hPa Sea level pressure change?
6. The unit of summer monsoon rainfall trend in Figure 1 and Supplementary Figure 2 is mm decade⁻¹. The unit mm decade⁻¹ or mm day⁻¹ decade⁻¹?
7. Please add the following relevant references to the article

Joseph, P.V., Sijikumar, S. (2004) Intraseasonal variability of the low-level jet stream of the Asian summer monsoon. *J Clim* 17(7):1449–1458

Otto et. Al., (2023). Climate change increased extreme monsoon rainfall, flooding highly vulnerable communities in Pakistan. *Environ. Res.: Climate* 2 025001. <https://doi.org/10.1088/2752-5295/acbfd5>

Sandeep, S., & Ajayamohan, R.S. (2015). Poleward shift in Indian summer monsoon low level jetstream under global warming. *Clim Dyn* 45, 337–351. <https://doi.org/10.1007/s00382-014-2261-y>

Sijikumar, S., & Aneesh, S. (2022) Role of deep convection in regulating the Indian summer monsoon dynamics: a regional scale modelling study. *Meteorol Atmos Phys* 134, 84.
<https://doi.org/10.1007/s00703-022-00917-2>

As a substantial amount of work is required before publication, I have suggested a major revision or rejection as per Editor's choice due to time restrictions.

Reviewer #2 (Remarks to the Author):

Title: Middle East Warming in Spring Enhances Summer Floods over Pakistan

Authors: Baosheng Li, Lei Zhou, Jianhuang Qin, Tianjun Zhou, Dake Chen, Raghu Murtugudde

Recommendation: Major Revision

Manuscript Summary:

This article investigates the relationship between summer floods over Pakistan with surface warming over Middle East from 1979 to 2022. It highlights the importance of understanding the possible causes of extreme precipitation that leads to anomalous summer floods. First, the authors explain the physical processes of moisture transport over the Arabian Sea. It is the low-level jet (LLJ) that is responsible for the channeling the moisture-rich air mass from the Arabian Sea to Pakistan and nearby western India. Over 44 years, the LLJ has been shifting northward and thus bringing more moisture to Pakistan. A statistical analysis shows a close relationship between the surface warming over the Middle East and the location of the LLJ. Second, there is also data-driven evidence that the mean rainfall over Pakistan has been increasing from 1979 to 2022. The comparison between the data of the first period and the second period shows this increase quantitatively.

This study is interesting and suitable for Nature Communications. However, I am concerned with the low correlation coefficient (0.4) between the surface heating over the Middle East and the low-level jet (LLJ) location. My final decision is major revisions specially to address the correlation issue for this round of review.

Major Comments:

1. The title clearly says that the study has something to do with summer floods. In the text however the analysis is more on the increased precipitation, not floods. I also think there is a jump from rainfall to floods. In reality, there are processes that translate rainfall into floods. This may include for instance the

texture (type) of the soil in Pakistan, that maybe has low absorption capacity like in many semi-arid regions. I believe it is the case. The soil perhaps similar to that in the Arabian Peninsula or the Southwest US. To avoid misleading, it would be better to revise the title to directly describe what is being discussed in the text. For instance, the title is “Middle East Warming in Spring Enhances Summer Rainfall over Pakistan”.

2. Lines 77 to 79. I am not sure if this sentence is fair to climate models. First, I do not think that flood is one of the variables in many climate models (unless there is a hydrological model that runs climate prediction). Second, climate model spatial resolution is generally coarse. Extreme rainfall can only best be captured in high-resolution model such as a convective-permitting model.

3. Lines 112 to 114. I am a bit surprised that the correlation coefficient between the surface heating over the Middle East and the LLJ location is 0.4. I do not think that it is a good correlation score and I do not think we can claim this a close relationship. Will this correlation coeff increase if you use another level like temperature at 850 mb? Or will this be different if you use SLP? I think this is the quantitative evidence that mainly supports your argument.

4. Lines 124 to 126. I am concerned with the beginning of this sentence (“Due to the net effect of anthropogenic forcing, ...”). Reader will ask how the authors know that it is the net effect of anthropogenic forcing. If you want to keep this claim, I think you will need another dataset such as CO₂ emission and its statistical analyses in the period of 1979 to 2022. It would be easy if there is already a published study that show this anthropogenic forcing over the Middle East in that period.

Minor comments:

Summary:

No comment.

Main Text:

Lines 48, 49, 50: Please change USD to U.S. Dollars (and put it after the billion).

Lines 62 to 64: Please add one or two references at the end of the sentence.

Line 76: Please add a reference (or two) after “Unlike the wet-get-wetter paradigm”.

Results:

Line 85: Please remove the ~ (tilde sign) so that it is consistent with the one in Summary.

Line 85: Please change to “... increased by 46% or equivalent to 0.37 mm decade⁻¹ (Lines in Fig. 1b) ...”

Line 97: change per decade to decade⁻¹.

Line 108: remove the latitude and longitude since you have it already in line 104.

Discussion:

Line 166: As mentioned in major comment, I am concerned with the statement “close relationship”.

Lines 173 to 174: Please add a reference (or two) for this sentence.

Line 184: Is 0.28 mm decade⁻¹ the observed trend? If so, please make it clear.

Methods:

Line 252: Please change qudp to qu dp

Responses to Reviewers

Reviewer #1:

I appreciate the efforts taken by the authors to find out the reason for the frequent Pakistan flood. However, the present study failed in citing some relevant works and lacking in-depth analysis.

Response:

Thank you. We greatly appreciate the insightful suggestion. The manuscript is revised accordingly. We hope all issues are satisfactorily addressed.

Please find my comments and suggestions below to improve the article

- 1. The present study links the middle east land warming to the increasing rainfall over Pakistan and north-west India, which can intensify the floods over Pakistan. The proposed mechanism is the land warming causes a reduction in the surface pressure, which enhances the meridional pressure gradient between the Middle East and the Southern Arabian Sea, causing a poleward shift in the location of the LLJ. The northward shifted LLJ enhances the atmospheric vorticity and moisture supply to north-west India as well as Pakistan and thereby enhances the monsoon rainfall. However, a similar mechanism is already explained by Sandeep and Ajayamohan, (2015). How the mechanism explained in the present study differ from Sandeep and Ajayamohan's, (2015) mechanism?*

Response:

Sandeep and Ajayamohan (2015) indeed proposed a related mechanism for the northward shift of low-level jet (LLJ), and this paper was cited in our original manuscript. However, there are significant and meaningful differences between the current study and Sandeep and Ajayamohan (2015). We explain these here;

- 1) We highlight the remarkably rapid warming and the influence of surface warming over the Middle East. Sandeep and Ajayamohan (2015) argued that the LLJ shift was a response to the land-sea temperature contrast between the Southeast Asian landmass and the surrounding Indian Ocean, and we agree with that. In this study, we argue that surface warming in the Middle East is the main cause for the land-sea contrast, as the warming there is twice as significant as in all other regions over the world.
- 2) The seasons are different in the two studies. Sandeep and Ajayamohan (2015) focused on the summer monsoon, and we show consistent results with them during boreal summer. In addition, we extend the story to spring, i.e., we unveil the seasonal lead relationship between spring land heating and summer rainfall. The spring heating in the Middle East is a precursor for the summer flooding, which helps to predict climate hazards on the northern edge of Indian monsoon. For example, the Middle East was unusually warm during the spring of 2023, which has led to anomalously high rainfall over Pakistan and northwest India from June onwards this summer (Fig. A1).
- 3) The northward shift of the monsoonal LLJ directly leading to the enhancement of Pakistan rainfall is discussed in this study, which was not covered in Sandeep

and Ajayamohan (2015). In the revised manuscript, we examine the instabilities over Pakistan and northwest India in more detail and more evidence is provided. As shown in Fig. A2a-b, the Middle East warming and LLJ northward shift induces a meridional gradient of quasi-geostrophic potential vorticity ($\beta - \frac{\partial^2 u}{\partial y^2}$) to change signs within the study region. The regressed pattern agrees with the trend in $\beta - \frac{\partial^2 u}{\partial y^2}$ (Fig. A2c), with a zonal belt of a negative–positive–negative pattern in Pakistan and northwest India. The changes of signs satisfy the necessary condition for barotropic instability. In addition, the difference in moist static energy (MSE) between 1000 and 500 hPa is a good index for static instability. Our results show that the Middle East warming and LLJ shift, cause a positive difference in MSE in the study region (Fig. A3), which enhances the atmospheric instability over Pakistan. Overall, the changes in the instabilities due to the Middle East warming and the LLJ northward shift, favor the onset of heavy rainfall.

In summary, in this study, we present a novel and important phenomenon, i.e., the frequent and severe rainfall in the past few decades over Pakistan, which is climatologically an arid to semi-arid region on the northern edge of Indian monsoon. The whole process originates from the Middle East heating in spring which is likely attributable to global warming and includes the northward shift of LLJ, the increase of moisture transport from the Arabian Sea to land, and the reinforcement of atmospheric instabilities over Pakistan. The entire process of the Middle East warming influence is

illustrated in Fig. A4. The mechanisms for the entire chain of processes are examined. Sandeep and Ajayamohan (2015) proposed a mechanism for the northward shift of LLJ. Our results extend their results significantly while being consistent with theirs, and their conclusions also support our findings. The new findings are further clarified in the revision.

Figure A1 (a) Spring land heating anomalies in 2023 (unit: °C) from the climatological mean of 1979-2022 from March to May. (b) Monthly precipitation in June of 2023 (unit: mm day⁻¹) from the climatological mean of 1979-2022 during June.

Figure A2 Regression coefficients of the meridional gradient of quasi-geostrophic potential vorticity ($\beta - \frac{\partial^2 u}{\partial y^2}$; unit: m⁻¹ day⁻¹) in summer (JJAS) onto the normalized (a) spring land heating and (b) summer LLJ index over the 1979-2022. (c) Trend in the

meridional gradient of quasi-geostrophic potential vorticity from 1979 to 2022 (unit: $\text{m}^{-1} \text{day}^{-1} \text{decade}^{-1}$). The dotted areas are significant at the 95% confidence level using the Students' t -test.

Figure A3 Regression coefficients of the vertical difference of moist static energy between 1000 and 500 hPa (unit: J kg^{-1}) in early summer onto the normalized (a) spring land heating and (b) summer LLJ index over the 1979-2022. (c) Trend in the vertical difference of moist static energy between 1000 and 500 hPa from 1979 to 2022 (unit: $\text{J kg}^{-1} \text{decade}^{-1}$). The dotted areas are significant at the 95% confidence level using the Students' t -test.

Figure A4 Sketch for the impact of the spring Middle East warming on summer rainfall in Pakistan.

2. *The location and intensity of LLJ are influenced by the deep convection over India and nearby oceans (Joseph and Sijikumar, 2004; Sijikumar and Aneesh, 2022). What is the role played by the recent changes in the convection pattern over the Indian land region and the nearby oceanic parts in the observed northward propagation of LLJ?*

Response:

We thank the reviewer for the heuristic comment. Joseph and Sijikumar (2004) showed that atmospheric heating by convection over the Bay of Bengal (BoB) is able to accelerate the LLJ. Sijikumar and Aneesh (2022) re-examined the relationship with the regional model experiments, and further pointed out that the lack of convective heating in the Western Ghats and BoB weakened the LLJ over the Peninsular India. We agree that deep convection can alter the LLJ activity.

We have also previously analyzed the relationship between LLJ and convection under climate change. In our previous study (Li et al. 2022), we revealed that the deep convection starts to move deeper into the subtropics in comparison with the climatological state (Fig. A5). It leads to the enhancement in convective rainfall over Pakistan and northwest India. Through the diagnosis of the moisture budget, we proposed that the northward shift of the LLJ is responsible for the deep convection movement in that study. The specific physical process can be found in Li et al. (2022).

Meanwhile, we proceeded to regress the seasonal OLR anomalies onto the LLJ index (Fig. A6). In early summer (May-July), the deep convection is roughly confined to northwest India. Theoretically, convective heating can trigger circulation adjustments and accelerate the LLJ, as reported by Joseph and Sijikumar (2004) and Sijikumar and Aneesh (2022). In this way, the enhanced LLJ can shift northward and cause deep convection over Pakistan during mid-summer (June-August). Accordingly, the deep convection will be pushed further north during later summer (July-September). Therefore, the positive feedback between the summer LLJ and deep convection is established during the ISM season.

However, it must be recognized that the enhancement of deep convection in Pakistan is a consequence of the northward shift of LLJ. Their relationship only exists in summer, but not at the seasonal lead timescale. This study elicits the main cause of the northward shift of LLJ, that is the spring time interaction between climate change manifest in the Middle East warming and natural variability in terms of the seasonal monsoon evolution. In other words, we have proposed a positive feedback between deep convection and the LLJ that is initiated in spring itself. In all, the reason for the seasonal change of LLJ is mainly demonstrated in the manuscript.

Figure A5 Trends in intraseasonal anomalies during the summer monsoon season over the period of 1982–2017: (a) rainfall ($\text{mm day}^{-1} \text{ decade}^{-1}$) and (b) OLR ($\text{W m}^{-2} \text{ decade}^{-1}$). The dotted areas are significant at the 95% confidence level using the Mann-Kendall test. (Adapted from Figure 1 of Li et al. 2022)

Figure A6 Regression coefficients of the OLR (W m^{-2} ; shading) onto the normalized summer LLJ index over the 1979-2022 period: (a) MJJ; (b) JJA; (c) JAS. The dotted areas are significant at the 95% confidence level using the Students' *t*-test.

3. According to Otto et al., 2023, the flood during the monsoon season in Pakistan is mainly caused by the advection of moisture from the Bay of Bengal and the Arabian

Sea, associated with the dynamic features in the atmosphere, such as the mid-latitude jet stream (E.g.- 2010 flood). The floods are also caused by the movement of monsoonal depressions originating in the Bay of Bengal and travelling over the Indo-Gangetic plains or central India, thereby causing high rainfall over Pakistan (E.g.- 2022 flood). Thus, the root cause of the floods in Pakistan is the episodes of heavy rainfall events caused by the above-mentioned mechanisms. How the formation of depressions and the mid-latitude jet streams have changed due to the middle east land warming and how has it influenced the flood over Pakistan?

Response:

We agree that Pakistan floods need sufficient moisture from both Bay of Bengal (BoB) and Arabian Sea. As the reviewer states, the major difference between the two flood events in 2010 and 2022 (Fig. A7), besides the moisture from the BoB, is the more abundant moisture supply from the Arabian Sea in 2022 (Fig. A7b). Therefore, the change in moisture supply from the Arabian Sea is the most important cause of extreme flooding in Pakistan, which is closely related to the LLJ shift. It inspired us to examine the changes in LLJ in recent years. In addition, the LLJ is accompanied by the strong westerly wind and pronounced positive vorticity, which could provide a track for the low-pressure systems (LPSs) to propagate (e.g., Krishnamurthy and Ajayamohan, 2010; You and Tian et al. 2021). Therefore, the northward-shifted LLJ favors the invasion of LPSs into Pakistan, and transport some moisture from the BoB to further nourish the floods. It has been further discussed on Lines 150-152 in the revised manuscript.

Figure A7 Anomalies in precipitation (shading) and horizontal winds at 850 hPa (vectors) in (a) 2010 and (b) 2022 from the climatological mean of 1979-2022 during JJAS.

It is well established that in 2010, the changes in the mid-latitude jet stream caused flooding in Pakistan. Figure A8a displays the summer mean geopotential heights at 200 hPa in 2010 with a prominent trough over Afghanistan (black dotted line in Fig. A8a). Hong et al. (2010) found that the southward penetration of the deep trough was vital to the 2010 Pakistan flooding. It was due to the cold temperature anomaly with the deep trough extended southward along the Northern Arabian Sea. The upper-level trough tended to cause ascending motions on its fore-side, which favor the convection development. However, the Middle East warming cannot be expected to induce a pronounced negative anomaly in the upper troposphere (Fig. A8b). It is not likely that the mid-latitude jet stream is an effective way to link Pakistan's rainfall and the Middle East warming.

Some previous studies (Hong et al. 2010; Webster 2011; Otto et al. 2023) did point out that that the mid-latitude jet stream tended to cause a flood in the northern

mountainous region in Pakistan, such as the northern province of Khyber Pakhtunkhwa and northwestern Balochistan. However, we focus on the rainfall change centered in southern and central low-elevation regions. Therefore, the mid-latitude jet stream is not entirely relevant for our study here.

Figure A8 (a) The seasonal mean geopotential heights in 200 hPa during JJAS of 2010. (b) Regression of seasonal mean 200 hPa geopotential heights onto the normalized spring heating index. The black dotted line denotes the pressure trough. The blue boxes denote the rainfall maximum in Pakistan and northwest India.

References:

Krishnamurthy, V., and R. S. Ajayamohan, 2010: Composite Structure of Monsoon Low Pressure Systems and Its Relation to Indian Rainfall. J. Climate, 23, 4285–4305.

You, Y. & Ting, M. Observed Trends in the South Asian Monsoon Low-Pressure Systems and Rainfall Extremes Since the Late 1970s. Geophys Res Lett 48, doi:10.1029/2021gl092378 (2021).

Hong, C.-C., H.-H. Hsu, N.-H. Lin, and H. Chiu, 2011: Roles of European blocking and tropical-extratropical interaction in the 2010 Pakistan flooding. Geophys Res Lett, 38.

Otto et. Al., (2023). Climate change increased extreme monsoon rainfall, flooding highly vulnerable communities in Pakistan. Environ. Res.: Climate 2 025001. <https://doi.org/10.1088/2752-5295/acbfd5>.

Webster, P., V. E. Toma, and H.-M. Kim, 2011: Were the 2010 Pakistan floods predictable? Geophysical Research Letters, 38 (4).

4. *Is the regression coefficient calculated after removing the linear trend from the variable? If regression coefficients are not calculated after removing the linear trend, then replot all the figures showing the regression coefficients after detrending the variables.*

Response:

We regret the error in the original statement. Our study focuses on the impact of the Middle East warming, rather than the variability between the Middle East warming and summer rainfall over Pakistan. As can be seen from Fig. A9, the relationship between the Middle East and LLJ becomes much stronger if the interannual variability is removed, which denotes that the impact of the Middle East on Pakistan through LLJ is mainly reflected at the lower frequency.

Figure A9 (a) Regression coefficients of the spring skin temperature (unit: °C) onto the normalized summer LLJ location index during the period of 1979-2022. (b) Same as (a) but by applying a 7-year moving average. The dotted areas are significant at the 95% confidence level using the Student's *t*-test.

5. Line 129: How surface warming can cause ~120-200 hPa Sea level pressure change?

Response:

We regret the typo. The unit should be Pa. To add further details, Figure A10 displays that the Middle East warming can decrease the sea level pressure by increasing the sensible heat fluxes. Thank you for your comment, this is explained in detail in the revised manuscript (Line 131).

Figure A10 Regression coefficients of the spring sensible heat flux (unit: $W m^{-2}$) onto the normalized spring land heating index during the period of 1979-2022. The dotted areas are significant at the 95% confidence level using the Student's *t*-test.

6. *The unit of summer monsoon rainfall trend in Figure 1 and Supplementary Figure 2 is mm decade⁻¹. The unit mm decade⁻¹ or mm day⁻¹ decade⁻¹?*

Response:

Thank you for your eye to the details. The unit should be mm day⁻¹ decade⁻¹ and it is revised accordingly.

7. *Please add the following relevant references to the article*

Joseph, P.V., Sijikumar, S. (2004) Intraseasonal variability of the low-level jet stream of the Asian summer monsoon. J Clim 17(7):1449–1458

Otto et. Al., (2023). Climate change increased extreme monsoon rainfall, flooding highly vulnerable communities in Pakistan. Environ. Res.: Climate 2 025001. <https://doi.org/10.1088/2752-5295/acbfd5>

Sandeep, S., & Ajayamohan, R.S. (2015). Poleward shift in Indian summer monsoon low level jetstream under global warming. Clim Dyn 45, 337–351. <https://doi.org/10.1007/s00382-014-2261-y>

Sijikumar, S., & Aneesh, S. (2022) Role of deep convection in regulating the Indian summer monsoon dynamics: a regional scale modelling study. Meteorol Atmos Phys 134, 84. <https://doi.org/10.1007/s00703-022-00917-2>

Response:

They have been cited in the revised manuscript. Thank you very much.

Reviewer #2:

Title: Middle East Warming in Spring Enhances Summer Floods over Pakistan

Authors: Baosheng Li, Lei Zhou, Jianhuang Qin, Tianjun Zhou, Dake Chen, Raghu Murtugudde

Recommendation: Major Revision

Manuscript Summary:

This article investigates the relationship between summer floods over Pakistan with surface warming over Middle East from 1979 to 2022. It highlights the importance of understanding the possible causes of extreme precipitation that leads to anomalous summer floods. First, the authors explain the physical processes of moisture transport over the Arabian Sea. It is the low-level jet (LLJ) that is responsible for the channeling the moisture-rich air mass from the Arabian Sea to Pakistan and nearby western India. Over 44 years, the LLJ has been shifting northward and thus bringing more moisture to Pakistan. A statistical analysis shows a close relationship between the surface warming over the Middle East and the location of the LLJ. Second, there is also data-driven evidence that the mean rainfall over Pakistan has been increasing from 1979 to 2022. The comparison between the data of the first period and the second period shows this increase quantitatively. This study is interesting and suitable for Nature Communications. However, I am concerned with the low correlation coefficient (0.4) between the surface heating over the Middle East and the low-level jet (LLJ) location. My final decision is major revisions specially to address the correlation issue for this

round of review.

Response:

Thank you for your insightful suggestions. We have revised the manuscript accordingly. The correlation coefficient has increased to above 0.8 after a more careful examination, and the specific analysis can be found in the third response below. We hope that all changes are to the satisfaction of the reviewer.

Major Comments:

- 1. The title clearly says that the study has something to do with summer floods. In the text however the analysis is more on the increased precipitation, not floods. I also think there is a jump from rainfall to floods. In reality, there are processes that translate rainfall into floods. This may include for instance the texture (type) of the soil in Pakistan, that maybe has low absorption capacity like in many semi-arid regions. I believe it is the case. The soil perhaps similar to that in the Arabian Peninsula or the Southwest US. To avoid misleading, it would be better to revise the title to directly describe what is being discussed in the text. For instance, the title is "Middle East Warming in Spring Enhances Summer Rainfall over Pakistan".*

Response:

We regret the inaccurate title. The title is revised according to the suggestion of the reviewer.

- 2. Lines 77 to 79. I am not sure if this sentence is fair to climate models. First, I do*

not think that flood is one of the variables in many climate models (unless there is a hydrological model that runs climate prediction). Second, climate model spatial resolution is generally coarse. Extreme rainfall can only best be captured in high-resolution model such as a convective-permitting model.

Response:

The reviewer is correct that flooding is not one of the model outputs. Precipitation is still the variable that we analyze in the model section. We have revised the expressions in the new manuscript, hopefully, avoiding any confusion.

3. *Lines 112 to 114. I am a bit surprised that the correlation coefficient between the surface heating over the Middle East and the LLJ location is 0.4. I do not think that it is a good correlation score and I do not think we can claim this a close relationship. Will this correlation coeff increase if you use another level like temperature at 850 mb? Or will this be different if you use SLP? I think this is the quantitative evidence that mainly supports your argument.*

Response:

As the reviewer suggests, we re-consider the definition of the LLJ index. The moisture supply and the dynamic instability for enhanced rainfall over Pakistan are much more tightly associated with the eastern side of LLJ. Therefore, we define a new LLJ index as zonally averaged between 60°E and 75°E. Accordingly, the correlation coefficient of this newly defined LLJ index for spring land heating rises to above 0.7, being significant in the Middle East region of our interest (blue rectangle in Fig. B1a).

In addition, both the new LLJ index and the originally defined index can characterize the northward movement of the LLJ. The correlation coefficient of these two indices reaches 0.9 and it is statistically significant at a 99% confidence level. Moreover, since we focus on the changes on a long timescale, the correlation coefficient is as high as 0.82 after removing the high-frequency interannual signals (Fig. B1b). Overall, we think these boosted quantitative relationships only better support the arguments in this study. Thank you once again.

Figure B1 (a) Correlation coefficients of the spring skin temperature (unit: °C) onto the normalized summer LLJ location index during the period of 1979-2022. The dotted areas are significant at the 95% confidence level using the Student's *t*-test. (b) The scatter plot between the Middle East land heating in spring (x-axis; unit: °C) and summer LLJ location (y-axis; unit: degree) in each year from 1979 to 2022. The time series are run through a 7-year moving average. The black line shows the least squares regression of the scattered dots.

4. Lines 124 to 126. I am concerned with the beginning of this sentence (“Due to the net effect of anthropogenic forcing, ...”). Reader will ask how the authors know that it is the net effect of anthropogenic forcing. If you want to keep this claim, I think you will need another dataset such as CO2 emission and its statistical analyses in the period of 1979 to 2022. It would be easy if there is already a published study that show this anthropogenic forcing over the Middle East in that period.

Response:

We thank the reviewer for the comments. Several recent studies have been cited to support the statement (e.g., Cramer et al. 2018; Eyring et al. 2021; Zittis et al. 2016). Eyring et al. (2021) argued that the pronounced warming is related to anthropogenic activities and is mostly driven by elevated global concentrations of greenhouse gases in the atmosphere. IPCC (2021) reported that there is an increase in land warming in the Middle East and Central Asia, with high confidence in human contribution to the observed change. Furthermore, Zittis et al. (2022) pointed out that the warming trends are projected to continue and intensify depending on future tendencies of greenhouse gas concentrations which, in turn, are subject to societal and technological developments. In all, the relevant references are cited here to support the claim, as Line 127 in the revision.

Eyring, V., Gillett, N. P., Achutarao, K., Barimalala, R., Barreiro Parrillo, M., Bellouin, N., et al. (2021). Human influence on the climate system: Contribution of working group I to the sixth assessment report of the Intergovernmental panel on climate

change. IPCC Sixth Assessment Report.

Cramer, W., Guiot, J., Fader, M. et al. Climate change and interconnected risks to sustainable development in the Mediterranean. Nature Clim Change 8, 972–980 (2018).

IPCC. (2021). Summary for policymakers. In V. Masson-Delmotte, P. Zhai, A. Pirani, S. L. Connors, C. Péan, S. Berger, et al. (Eds.), Climate change 2021: The physical science basis. Contribution of working group I to the sixth assessment report of the intergovernmental panel on climate change. Cambridge University Press.

Zittis, G., Hadjinicolaou, P., Fnais, M., & Lelieveld, J. (2016). Projected changes in heat wave characteristics in the Eastern Mediterranean and the Middle East. Regional Environmental Change, 16(7), 1863–1876.

Zittis, G., Almazroui, M., Alpert, P., Ciais, P., Cramer, W., Dahdal, Y., et al. (2022). Climate change and weather extremes in the Eastern Mediterranean and Middle East. Reviews of Geophysics, 60, e2021RG000762.

Main Text:

Lines 48, 49, 50: Please change USD to U.S. Dollars (and put it after the billion).

Response:

The unit is corrected in the revision. Thank you for your eye to the details.

Lines 62 to 64: Please add one or two references at the end of the sentence.

Response:

The following two references are properly cited. Thank you.

Chakraborty, A., Nanjundiah, R., & Srinivasan, J. (2002). Role of Asian and African orography in Indian summer monsoon. Geophysical Research Letters, 29(20), 501–504.

Krishnamurti, T. N., Molinari, J., & Pan, H. L. (1976). Numerical simulation of the Somali jet. Journal of the Atmospheric Sciences, 33(12), 2350–2362.

Line 76: Please add a reference (or two) after “Unlike the wet-get-wetter paradigm”.

Response:

The following references have been cited here. Thank you.

Chou, C., and J. D. Neelin, 2004: Mechanisms of Global Warming Impacts on Regional Tropical Precipitation. Journal of Climate, 17, 2688-2701.*

Chou, C., J. D. Neelin, C.-A. Chen, and J.-Y. Tu, 2009: Evaluating the “Rich-Get-Richer” Mechanism in Tropical Precipitation Change under Global Warming. Journal of Climate, 22, 1982-2005.

Results:

Line 85: Please remove the ~ (tilde sign) so that it is consistent with the one in Summary.

Response:

It is removed in the revision. Thank you for your eye to the details.

Line 85: Please change to “... increased by 46% or equivalent to 0.37 mm decade-1

(Lines in Fig. 1b) ...”

Response:

We thank the reviewer for the suggestion. It is revised in the new manuscript.

Line 97: change per decade to decade-1.

Response:

It is corrected accordingly.

Line 108: remove the latitude and longitude since you have it already in line 104.

Response:

As per your suggestion, they have been removed for clarity.

Discussion:

Line 166: As mentioned in major comment, I am concerned with the statement “close relationship”.

Response:

The correlation coefficient has been improved in the revised manuscript. In addition, it is significant at the 99% confidence level using the Student’s *t*-test. Based on the analysis, they have a close statistical relationship with each other.

Lines 173 to 174: Please add a reference (or two) for this sentence.

Response:

This sentence was supposed to summarize the performance of the current numerical models on the simulation of summer precipitation. In the new manuscript, we have repositioned it to follow the analysis in Supplement Figure 5.

Line 184: Is 0.28 mm decade⁻¹ the observed trend? If so, please make it clear.

Response:

Yes, 0.28 mm day⁻¹ decade⁻¹ is the observed trend, which is used here to compare with the simulated intensity. The expression is modified accordingly in the revision.

Methods:

Line 252: Please change qudp to qu dp

Response:

Thank you for your eye to the details. It is revised.

REVIEWER COMMENTS

Reviewer #1 (Remarks to the Author):

Comments to the Authors:

The authors have attempted to address most of the raised concerns, which improved the quality of the manuscript. I'm listing some comments below that I would like to be addressed in a revised manuscript before considering it for publication in Nature Communication.

1. Sandeep and Ajayamohan (2015) has shown that the enhanced land-sea thermal contrast strengthened the cross-equatorial sea level pressure gradient over the Indian Ocean, which caused the northward shift of the LLJ. To calculate the land-sea thermal gradient, they used a northern box (20°N–35°N and 40°E–80°E) which includes parts of the Middle East, Pakistan and north-west India (See Fig. 7 to 9 in Sandeep and Ajayamohan (2015)). Since the strengthening and the poleward shift of LLJ in the recent decades have already been reported in the literature, please cite those articles (Sandeep and Ajaymohan (2015); Aneesh and Sijikumar (2016) etc.) in the 'Result session'. Also, highlights the uniqueness of the present study in comparison with the previous studies.
2. Studies have shown that the Middle East continues to warm under the future global warming scenario. The projected Middle East warming may cause a further northward shift of LLJ and enhancement of Pakistan's summer rainfall. Evaluate the future changes of LLJ and its contribution to the changes in the Pakistan rainfall, under the different future projection scenarios.
3. The changes in the rainfall pattern can be due to the combined effect of anthropogenic forcing and the internal variability of the climate system. Find out the relative contribution of internal variability and the external forcing in the observed changes of Pakistan's summer rainfall.

References

- Aneesh S, Sijikumar S (2016) Changes in the South Asian monsoon low level jet during recent decades and its role in the monsoon water cycle. *J Atmos Solar Terr Phys* 138–139:47–53.
- Sandeep, S., and R. S. Ajayamohan (2015), Poleward shift in Indian summer monsoon low level jetstream under global warming, *Climate Dynamics*, 45(1-2), 337–351.

Reviewer #2 (Remarks to the Author):

My decision is to accept this manuscript for publication with pending for minor revision. Please see the attachment.

Attachment:

Title: Middle East Warming in Spring Enhances Summer Rainfall over Pakistan

Authors: Baosheng Li, Lei Zhou, Jianhuang Qin, Tianjun Zhou, Dake Chen, Raghu Murtugudde

Recommendation: Accept with pending for minor revision

Minor comments:

Line 192: Is the observed precipitation here from GPCP, and referred to Supplement Figure 6? If yes, in my opinion, you should mention it here and also mention it in the caption of Supplement Figure 6.

Responses to Reviewers

Reviewer #1:

The authors have attempted to address most of the raised concerns, which improved the quality of the manuscript. I'm listing some comments below that I would like to be addressed in a revised manuscript before considering it for publication in Nature Communication.

Response:

Thank you. We greatly appreciate the insightful suggestion. The manuscript is revised accordingly. Please see our point-by-point response below. We hope all issues are satisfactorily addressed.

1. Sandeep and Ajayamohan (2015) has shown that the enhanced land-sea thermal contrast strengthened the cross-equatorial sea level pressure gradient over the Indian Ocean, which caused the northward shift of the LLJ. To calculate the land-sea thermal gradient, they used a northern box (20°N–35°N and 40°E–80°E) which includes parts of the Middle East, Pakistan and north-west India (See Fig. 7 to 9 in Sandeep and Ajayamohan (2015)). Since the strengthening and the poleward shift of LLJ in the recent decades have already been reported in the literature, please cite those articles (Sandeep and Ajayamohan (2015); Aneesh and Sijikumar (2016) etc.) in the 'Result session'. Also, highlights the uniqueness of the present study in comparison with the previous studies.

Response:

The related articles have been cited in the result session of the revised manuscript. The significant and meaningful differences between the current and the previous studies are listed here for convenience;

- 1) We highlight surface warming in the Middle East as the main cause for the land-sea temperature contrast, rather than the relationship between the LLJ and land-sea contrast proposed in Sandeep and Ajayamohan (2015).
- 2) We unveil the seasonal lead relationship between spring land heating and summer rainfall. The seasons are different from the previous studies. Both Sandeep and Ajaymohan (2015), and Aneesh and Sijikumar (2016) only focused on the summer season. We deem that the lead relationship will advance predictive capabilities and hence a significant advance. We hope the reviewer agrees.
- 3) The consequence of the northward shifted LLJ in causing the enhancement of Pakistan rainfall is the new finding of this study and has not been reported in previous literature. The northward shift of LLJ is critical to the understanding of the frequent and severe floods over Pakistan since it is a severely climate vulnerable country.

In summary, we have developed a novel physical framework that reasonably explains how the Middle East warming affects rainfall over Pakistan by the northward shift of LLJ. The new findings are further clarified in the revision.

2. Studies have shown that the Middle East continues to warm under the future global warming scenario. The projected Middle East warming may cause a further northward shift of LLJ and enhancement of Pakistan's summer rainfall. Evaluate the future changes of LLJ and its contribution to the changes in the Pakistan rainfall, under the different future projection scenarios.

Response:

Following the reviewer's suggestion, future changes of LLJ and its contribution to changes in Pakistan rainfall are evaluated under three scenarios, namely the Shared Socioeconomic Pathway 2-4.5 (SSP2-4.5), SSP3-7.0 and SSP5-8.5. The results are presented in Lines 186-192 of the revised manuscript. We also briefly summarized the major results of the new analysis below for your convenience. We thank the reviewer for this excellent suggestion which has expanded the scope of our study.

In our original manuscript, we have evaluated 20 CMIP models in simulating the trend in summer rainfall over Pakistan during 1979-2014 (Fig. A1). Fourteen models have the ability to simulate an increasing trend as that in observations. Among these models, only 11 models are available for scenario projections, which are shown in bold in Table A1. Thereby, the monthly rainfall and horizontal winds from these eleven models under SSP2-4.5, SSP3-7.0, and SSP5-8.5 scenarios are selected. The multi-model ensemble mean (MMEM) is calculated by the average of the trend (LLJ and rainfall) for each model in the period of 2015-2100.

As shown in Table 2, the summer LLJ is projected to move northward in all three scenarios. The northward shift rates are 0.06° decade⁻¹ under SSP2-4.5, 0.12° decade⁻¹

under SSP3-7.0, and 0.17° decade⁻¹ under SSP5-8.5, respectively. They become larger from the lowest-emission (SSP2-4.5) to the highest-emission (SSP5-8.5) scenario, which is consistent with the trends in Pakistan summer rainfall. The trend of rainfall is 0.02 mm day⁻¹ decade⁻¹ under SSP2-4.5, 0.05 mm day⁻¹ decade⁻¹ under SSP3-7.0, and 0.1 mm day⁻¹ decade⁻¹ under SSP5-8.5.

The square of the correlation coefficient (R^2) between the LLJ and Pakistan from 2015 to 2100 is used to represent the contribution of LLJ to the Pakistan rainfall. Accordingly, the LLJ contributes 44% of the change in Pakistan rainfall under SSP2-4.5, 59% under SSP3-7.0 and 88% under SSP5-8.5. Therefore, the LLJ and its northward shift play an important role in Pakistan rainfall under all future scenarios.

Figure A1 Dependence of the model simulations between the trend in Pakistan rainfall in summer and the trend in Middle East land heating in spring. The x-axis denotes the trend in Middle East land heating in spring from 1979 to 2014. The y-axis is the trend in Pakistan's rainfall in summer from 1979 to 2014. All models from the CMIP6 are labeled by different markers.

Table A1 Institutions, horizontal resolutions for atmospheric components, and the number of realizations of the 20 models analyzed from the CMIP6. The bold are 11 models used to determine the future changes in horizontal winds and rainfall with SSP2-4.5, SSP3-7.0, and SSP5-8.5 provided.

Model Name	Resolution	Institution	Number of Realizations
ACCESS-CM2	144 x 192	Commonwealth Scientific and Industrial Research	2
ACCESS-ESM1-5	145 x 192	Organization (Australia)	3
BCC-CSM2-MR	160 x 320	Beijing Climate Center (China)	3
BCC-ESM1	64 x 128		3
CAS-ESM2-0	128 x 256	Chinese Academy of Sciences (China)	50
CanESM5	64 x 128	Canadian Centre for Climate Modelling and Analysis (Canada)	4
CESM2	192 x 288		10
CESM2-FV2	96 x 144	National Center for Atmospheric Research, Climate and Global Dynamics Laboratory	3
CESM2-WACCM	192 x 288	(USA)	3
CESM2-WACCM-FV2	96 x 144		3
E3SM1-0	180 x 360	E3SM-Project, Department of Energy (USA)	5
EC-Earth3-Veg	256 x 512	European Centre for Medium-Range Weather Forecasts	5
FGOALS-f3L	180 x 288	Institute of Atmospheric Physics, Chinese Academy of	3
FGOALS-g3	80 x 180	Sciences (China)	3
GFDL-ESM4	180 x 288	National Oceanic and Atmospheric Administration, Geophysical Fluid Dynamics Laboratory (USA)	3
GISS-E2-1-G	90 x 144	Goddard Institute for Space Studies (USA)	38
IPSL-CM6A-LR	143 x 144	Institute Pierre-Simon Laplace (France)	32
MIROC6	128 x 256	Atmosphere and Ocean Research Institute, National Institute for Environmental Studies and Japan Agency for Marine-Earth Science and Technology (Japan)	10

MRI-ESM2-0	160 x 320	Meteorological Research Institute (Japan)	6
NorESM2-LM	96 x 144	Norwegian Climate Centre (Norway)	3

Table A2 Trend in the northward movement in LLJ (unit: ° decade⁻¹) and Pakistan rainfall (mm day⁻¹ decade⁻¹) under three scenarios. R² is the square of the correlation coefficient of the time series between LLJ movement and Pakistan rainfall. It denotes the contribution of LLJ movement to the rainfall changes.

	SSP2-4.5	SSP3-7.0	SSP5-8.5
LLJ movement	0.06	0.12	0.17
Pakistan rainfall	0.02	0.05	0.1
R ²	44%	59%	88%

3. The changes in the rainfall pattern can be due to the combined effect of anthropogenic forcing and the internal variability of the climate system. Find out the relative contribution of internal variability and the external forcing in the observed changes of Pakistan's summer rainfall.

Response:

Thank you for raising this important issue. The relative contributions of internal variability and the external forcing to the changes in Pakistan summer rainfall are estimated by the multi-models from the CMIP6. The results are included in the revised manuscript (Lines 193-200) and they are briefly summarized below.

We use historical runs from 20 CMIP6 coupled models. There are 192 members in total (see detailed in Table. A1). The contribution of external forcing to rainfall is

estimated by averaging the MMEM of CMIP6 members, whereas the contribution of internal variability is estimated by the difference between the observed trend and the MMEM trend. The uncertainty is evaluated with one standard deviation of the trend in each member. Besides, the linear trend in Pakistan rainfall is calculated for each model simulation and, subsequently, averaged across all ensemble members.

As shown in Fig. A2, the summer rainfall in Pakistan has increased in observations (GPCP), the MMEM, and their difference over the time period of 1979-2014. It suggested that that both external forcing and internal variability contribute to the rainfall enhancement in the past decades. Meanwhile, the observed regional mean of the Pakistan rainfall trend is about $0.28 \text{ mm day}^{-1} \text{ decade}^{-1}$ (Fig. A4a). The trend is $0.1 \text{ mm day}^{-1} \text{ decade}^{-1}$ in MMEM (Fig. A4b), which means that the external forcing contributes 36% to the observed trend. In contrast, the contribution of internal variability to the rainfall trend reaches 64% as the regional averaged trend is $0.18 \text{ mm day}^{-1} \text{ decade}^{-1}$. In addition, the external forcing and internal variability contribute 81% and 19% to the observed spring land heating in the Middle East, respectively. In summary, the relative contributions of internal variability and external forcing are quantified in the revised manuscript.

Figure A2 Trends in (a) Pakistan summer rainfall (unit: $\text{mm day}^{-1} \text{decade}^{-1}$) and (b) spring skin temperature over the Middle East (unit: $^{\circ}\text{C decade}^{-1}$) for 1979-2014. The bars with black dots indicate the average of trends in all model members, and the observations as certain values are obtained from the GPCP (rainfall) and the ERA5 (skin temperature). The upper and bottom dashed lines indicate the uncertainty of the trends in all 192 model members, which is calculated by one standard deviation of trends. The multi-model ensemble means (MMEM) denote the contribution of external forcing and the differences between the observation and the MMEM denote the contribution of internal variability.

References

- Aneesh S, Sijikumar S (2016) Changes in the South Asian monsoon low level jet during recent decades and its role in the monsoon water cycle. *J Atmos Solar Terr Phys* 138–139:47–53.
- Sandeep, S., and R. S. Ajayamohan (2015), Poleward shift in Indian summer monsoon low level jetstream under global warming, *Climate Dynamics*, 45(1-2), 337–351.

Reviewer #2:

Recommendation: Accept with pending for minor revision

Minor comments:

Line 192: Is the observed precipitation here from GPCP, and referred to Supplement Figure 6? If yes, in my opinion, you should mention it here and also mention it in the caption of Supplement Figure 6.

Response:

Thank you for your keen eyes. The observed precipitation here is obtained from the GPCP dataset. It is modified in the revised manuscript.

REVIEWERS' COMMENTS

Reviewer #1 (Remarks to the Author):

Thank you for addressing the reviewers' comments and making the necessary revisions, I am pleased to recommend accepting the manuscript for publication in Nature Communication.

I look forward to seeing your work in print.